# A Statistical Model to Estimate the Local Vulnerability to Severe Weather

Tobias Pardowitz[1,2]

[1] Hans Ertel Centre for Weather Research, Optimal Use of Weather Forecast Branch

[2] Freie Universität Berlin, Institute of Meteorology, Carl-Heinrich-Becker Weg 6-10, 12165 Berlin, Germany

*Correspondence to*: Tobias Pardowitz (tobias.pardowitz@met.fu-berlin.de)

**Abstract.** We present a spatial analysis of weather related fire brigade operations in Berlin. By comparing operation occurrences to insured losses for a set of severe weather events we demonstrate the representativeness and usefulness of such data in the analysis of weather impacts on local scales. We investigate factors influencing the local rate of operation occurrence.

While depending on multiple factors – which are often not available – we focus on publicly available quantities. These include topographic features, land use information based on satellite data and information on urban structure based on data from the open street map project. After identifying suitable predictors such as housing coverage or local density of the road network we set-up a statistical model to be able to predict the average occurrence frequency of local fire brigade operations. Such model can be used to determine potential "hotspots" for weather impacts even in areas or cities where no systematic records are

available and can thus serve as a basis for a broad range of tools or applications in emergency management and planning.

**Keywords.** Severe weather, Weather impacts, Vulnerability, Emergency management

## 1 Introduction

It has been stated within the Sendai Framework for Disaster Risk Reduction 2015-2030 by the United Nations (UNISDR,

2015) that the implementation of effective disaster risk reduction measures should be based on an understanding of disaster risks, including all its dimensions of vulnerability, capacity, exposure of persons and assets, hazard characteristics and the environment. On local and national levels, this requires to systematically evaluate, record, share and publicly account for disaster losses to gain understanding of the impacts in the context of event-specific hazard, exposure and vulnerability information.

While insurance records are a very useful data source and have been used in many analyses of regional weather impacts, their availability is generally limited due to economic interests of insurance providers. Making use of records of local emergency managers (first responders) yields an immense potential as an alternative database for analysing weather impacts, particularly on local scales. While often such records exist, they mostly lack systematic and homogenous data format and quality standards. Definition of such data standards must be regarded key requisite to be able to scientifically address disaster losses as required

within the Sendai Framework.

Relating emergency call data to extreme weather, most studies analyse ambulance callout data or emergency department visits in face of temperature extremes, in particular extreme heat (Bassil et al., 2005; Dolney and Sheridan 2008; Schaffer et al. 2012; Thornes et al. 2014). Wargon et al. (2009) have done a review on studies concerned with the modelling and forecasting of emergency department visits. It is found that the number of patient visits at emergency departments or walk-in clinics can be

modelled with rather good performance. Mostly based on predictors such as the day of the week or season these models explain between 31% and 75% of patient-volume variability. However, including meteorological data apparently failed to improve model performance (Wargon et al., 2009). Findings of more recent studies however do find that weather factors such as temperature and humidity play a role in the demand for ambulance services and demonstrate that including weather forecast data can in fact improve forecasts of daily ambulance demand (Wong and Lai, 2014).

There have been only few studies making use of spatial information of emergency callout data (i.e. the location of an assistance request) in relation to severe weather events. Two studies by Schuster et al. (2005) and Rossi et al. (2013) compared emergency call data with radar reflectivity data for a severe hailstorm event and found a satisfying representation of the hailstorm path in the density of emergency calls on the ground. Other studies tried to utilize similar data, however facing problems concerning the availability of accurate data. As described in Busch (2008), problems can occur in case of catastrophic events since the

archival of fire callout data is often limited in such cases. In particular, this means that spatial information on the individual location of callouts is not archived, hindering spatial analyses for these events.

Pardowitz and Göber (2016) have demonstrated -similar to the studies mentioned above- that satisfying correspondence of radar reflectivity for severe thunderstorm events and locations with occurrences of fire brigade operations can be found. However, this occurrence is strongly modulated by other factors such as the density of buildings. This is a confirmation of the

common understanding, that the occurrence and the height of impacts are determined by the simultaneous existence of a hazard and vulnerability against this hazard.

Approaches to address the local vulnerability have been developed in flood impact modelling. Apel et al. (2009) and Jongman et al. (2012) describe different modelling approaches to estimate economic damages for flood events (particularly the 2002 flood event in Saxony). Based on data from digital elevation models (DEM), local damages are estimated in dependence of

inundation depth. Furthermore, such depth-damage relation can be differentiated –e.g. by considering information on land use- to account for variable vulnerabilities.

In this study we focus exclusively on the estimation of predictors describing the local vulnerability and exposure. We thus neglect temporal variations of weather parameters and investigate the long-term averaged occurrence frequencies (which can be regarded as the equivalent to a local "climatology") of fire brigade operations. It is also tested in this study, whether it is

possible to predict these long-term occurrence frequencies for areas in which we might not have actual operation records available.

Based on a new dataset of fire brigade operations in Berlin for the period of 2002-2012, this study aims at assessing the latter, namely the vulnerability against hydro-meteorological hazards. In a first step we analyse this new dataset particularly with

respect to the question in how far these impacts are related to building damages induced by wind storms and thunderstorms. It needs to be noted that neither insurance loss data nor the archive of fire brigade operation extensively describes all possible weather impacts. Instead it is important to investigate on the different causes of weather impacts included in the individual datasets. Within the metropolitan area of Berlin, we then aim to identify factors describing the local vulnerability and thus influencing the local risk for weather impacts as given by the fire brigade operations. Potential factors include topographic features, land use information based on satellite data and information on urban structure based on data from the open street map project. After identifying suitable predictors such as housing density or local density of the road network we set-up a statistical model to be able to predict local operation densities. Such model can be used to determine potential "hotspots" for weather impacts even in areas or cities where no systematic records are available.

The remainder of this paper is structured as follows. Section 2 describes the various datasets that are used to describe impacts as well as potential predictors for vulnerability. Methodological steps and modelling approaches are described in Section 3, while results are shown in Section 4. Finally, Section 5 provides a discussion of results as well as the major conclusions that can be drawn from this study.

## 2 Data

### 2.1 Fire brigade operations

A dataset provided by the Berlin fire brigade is analysed, comprising weather related fire brigade operations for the period 2002-2011. The dataset contains location and time of alerts, as well as keywords associated to each operation indicating the type of operation. Keywords indicate "water-related" operations, "tree-related" operations, "traffic obstructions", operations related to "construction elements", operations due to "ice and snow" and few operations associated to other keywords. The keywords are assigned operationally by the Berlin fire brigade and reflect the type of (technical) operation the fire brigade had to handle. While "water-related" operations consist of flooded basements or other incidents requiring the disposal of water. the keyword "tree-related" refers to operations in which windthrow had to be handled. The keyword "traffic obstruction" embraces all operations dealing with the removal of obstacles to restore traffic while "construction element" refers to the removal of damages due to destroyed construction components. It needs to be noted that individual keywords might overlap in the sense that more than one keyword applies. However, for the present analysis we focus on the primary keywords assigned to an individual operation. Additional details on the usage of keywords of the Berlin fire brigade can be found in Kox et al. (2015). Total counts of weather related fire brigade operations in the period 2002-2011 accounted to slightly above 10.000 per year. This is about 27% of all operations of the Berlin fire brigade, which -according to the annual reports- accounted to about 37.000 operations per year in the same period. In comparison, fire extinction operations (about 7.500 per year) accounted for about 20% of all operations. Note that ambulance call outs (~245.000 per year) and false alarms (~31.000 per year) have been disregarded here. Most weather-related operations are due to water damages (33%). Traffic obstruction account for 25% of

operations and tree related for about 17% (Table 1). Operations related to construction elements accounted for about 14% and Ice and snow related operations for 2%. Some other keywords (individually accounting for 1% or less each) have been used which sum up to about 8%. Stratifying by season shows that in total, operations are equally distributed over winter (October-March) and summer half year (April-September). This choice is done primarily to best discriminate between thunderstorm and

winter storm impacts (compare e.g. Donat et al., 2011). The individual types of operations however partly show distinct differences in summer and winter (Table 1). Particularly tree related operations occur mainly in summer (73%) while ice and snow related operations naturally occur in winter exclusively.

## 2.2 Building Loss Data

Insurance data on windstorm and thunderstorm losses to residential buildings were provided by the German insurance association (Gesamtverband der Deutschen Versicherungswirtschaft e.V., GDV). Berlin-wide damages are available on daily basis for the period 1997-2011, while data on zip code level (190 within Berlin) is available for a small selection of events only. Direct (liquid) precipitation damages as well as flooding damages are not part of the available dataset even though they might be highly relevant in case of severe precipitation related to thunderstorm events. Still, for investigations of severe

weather events -particularly small-scale events such as thunderstorms- insurance loss data is thus extremely valuable. However, difficulties arise when interpreting the insurance data since the dataset does not allow a direct attribution of losses to their cause (i.e. hail or wind-storm induced). In addition, faulty attribution of individual insurance claims (both temporal and spatial) can cause inaccurate loss figures. E.g., this can be because the exact day of occurrence of a damage is unknown in some cases. In addition, if damage occurs at a house administered by a real estate management, the insurance claim might

be attributed according to their administrative centre instead of the actual origin. For the set of events for which insured loss data is available on zip code area an evaluation of the spatial patterns and a comparison to the occurrences of fire brigade operations can be made. The selection of events contains the 4 windstorm events with highest impacts in Berlin within the reference period 2007-2011 (Kyrill 17th-19th January 2007, Lothar07 24th-28th May 2007, Emma 28th February- 2nd March 2008, Xynthia 27th February – 1st March 2011) and 2 convective events (Aram 6th June 2011, Gunnar 22nd June 2011) that

have been selected since they were studied in detail in Wapler et al. (2015). The dataset comprises area-wide coverage of losses due to windstorm and thunderstorm. Also, to address the temporal correlation of different impacts, Berlin-wide losses (in €) are analysed and compared to total operation counts within Berlin. According to the insurance loss records (covering wind storm and hail damages), 8.12 Mio € damages are recorded for Berlin per year. The temporal distribution is rather balanced with 48% of damages occurring in summer and 52% in winter. While most damages in winter are related to intense

winter windstorms (Klawa and Ulbrich, 2003; Pinto et al, 2010; Donat et al., 2011), a large share of summer damages is due to thunderstorms and in particular due to hailfall (Aller and Kozlowski, 2008; Kunz and Puskeiler, 2010).

## 2.3 Open Street Map (OSM) Data

Data from the open source project OpenStreetMap (OSM, www.openstreetmap.org) are used to derive predictors for local vulnerability. Particularly we analyse georeferenced information on individual buildings (including their location and extent) as well as information on road networks. As a first predictor, the number of buildings per grid cell on a regular 1x1km grid is derived. Also, by including information on housing extent, the fraction of the grid cell covered by buildings is calculated. As discussed later, even though these quantities are highly correlated, both predictors are valuable to be considered since enabling the distinction between high density city centre with very large buildings in comparison to suburban areas with high numbers of detached houses. Additionally, the density of the road networks is considered by calculating the total length of road segments within a 1x1km grid cell (specified as a length per grid cell area, thus km/km²). The OSM dataset contains a classification of the road networks (the major categories being highway, primary, secondary and tertiary road networks), which is why road densities can be assessed individually for these classes. It would be valuable to add the population density as a predictor as well. However, population density is not freely available on the spatial resolution required in this analysis (1 km). Freely available datasets include the CIESIN global gridded dataset (with a resolution of about 5 km) or from the German Federal Statistical Office (DESTATIS) which is available on district level only.

## 2.4 CORINE land cover data

The CORINE (Coordination of Information on the Environment) land cover (CLC) data set provides European-wide information on land cover and land use, based on a unified classification of the most important types of land usage (CEC, 1994; Bossard et al., 2000, Büttner et al., 2012). More specifically, we used CLC2006, which is based on SPOT-4/5 and IRS P6 LISS III satellite data. Geometric accuracy of satellite images is specified to be smaller than 25m and resulting minimum mapping units within CLC are specified to be 25 ha with a geometric accuracy of the CLC data being better than 100m. In total, 44 land usage classes are used in CLC2006 being subcategories of the main land usage types "Artificial surfaces", "Agricultural areas", "Forest and semi natural areas", "Wetlands" and "Water bodies". More details on CLC2006 can be found in Büttner et al. (2012). The original data consists of polygon data in form of shape files, which have been processed to calculate land use characteristics on a grid-point basis. For this, the area fractions of all 44 CLC types (adding up to 100%) are calculated on a specified grid. Here we use a regular lon-lat grid with a 1x1km resolution. These gridded fields of the area fraction are then used as predictors in the following analyses.

## 2.5 Data from digital elevation model

Data from the digital elevation model dgm200 (GeoBasis-DE / BKG 2016) as well is used to derive orographic height and slope. Original data has a horizontal resolution of 200m and is available for the territory of Germany. Alternatively, GTOPO30 has been used which has a lower horizontal resolution of 30 arc seconds (approximately 1km). However, GTOPO30 is available globally. The data is used to derive orographic height as well as the slope on a regular 1x1km grid for Germany. In case of dgm200 which has a finer resolution compared to the target grid, orographic height is calculated as the average height over all original 200m x 200m grid boxes within a target grid box. In case of GTOPO30 orographic height on the target grid

is determined by means of a nearest neighbour remapping. Since differences for the Berlin region were negligible, dgm200 has been used in the following. The slope is calculated according to the algorithm proposed by Horn (1981). The algorithm also assesses the aspect, which –in further studies- might be considered as an additional vulnerability predictor. However, since the area for which the vulnerabilities are analyses is limited to Berlin (featuring no considerable height variations), topographic features play only a minor role and do not play a major role here. However, in future studies including other investigation areas, topographic features may be more important to consider.

## 3 Methodology

### 3.1 Comparison of fire brigade operations and building damage data

To assess in how far representative spatial information on weather impacts on a sub-city scale can be derived from the dataset and in how far there is a temporal correspondence between daily damages and operation numbers, a comparison of building loss data and fire brigade operations is performed. While insured losses on residential housing comprise specific impacts caused by windstorms and thunderstorms, fire brigade operations can be caused by additionally meteorological phenomena such as flooding (which is not included in the loss dataset) or impacts due to freezing rain or road icing. The aim of this comparison is to identify in how far specific categories of fire brigade operations (i.e. tree related operations) can be related to the wind and hail impacts as described by the insured loss data. It can be expected that other categories (operations due to road icing) will not relate to insured losses.

For a set of events, including 2 convectively driven summer events and 4 windstorm events, a qualitative and quantitative comparison is performed between the spatial patterns of building damages and the occurrence of fire brigade operations. This is done by calculating total operation count numbers for zip code areas (190 within Berlin) for each of the 6 events. Besides total operation numbers, counts for operations related to individual keywords are assessed. Resulting maps are compared and spatial correlations calculated. Spatial correlations (i.e. measuring the correlation of spatial variations amongst zip codes) are calculated using the Pearson correlation. Since it cannot be assumed that all considered parameters are gaussian distributed, it was tested in how far results differ using the Spearman rank correlation. It was found, that results are not qualitatively affected by the chosen correlation method. Correlations are tested for significance by testing whether the Pearson's product moment correlation follows a t distribution. Significance of correlations is assessed by considering the resulting p-values. Daily total operation counts for Berlin are furthermore compared to Berlin-wide damages, which are available on daily level for the period of 2002-2011. Temporal correlations to daily building damages are calculated, again for both total count and counts for operations related to individual keywords.

### 3.2 Spatial correlation between potential vulnerability predictors and patterns of operation occurrences

To identify predictors for vulnerability, a spatial correlation analysis between numerous quantities derived from the different geospatial data sets and gridded operation densities is performed. Variables include gridded densities of man-made structures

(buildings, streets), topographic features (height, slope) as well as land use information. The latter are pre-processed such that the area fraction of a specific land use type (as specified in the CORINE data set) within each 1x1km grid box is given. Again, spatial correlations are assessed using either operations of one specific category or operations irrespective of their type. As described above, correlations are calculated using the Pearson correlation. Again, using the Spearman rank correlation did not

qualitatively affected the results. Significance of correlations is assessed as described in the previous section.

### 3.3 Multiple linear regression model

On the basis of the set of potential vulnerability predictors (as listed in Table 3), a statistical model is set up based on multiple linear regression to analyse the predictability of the spatial distribution of (long-term) operation occurrence rates. Such model

could potentially be used to identify "hotspots" in the local occurrence of operations in areas where no explicit data on operations is available and might be highly relevant in terms of long term planning of capacities for an effective emergency management. In the following, three different types of models are addressed. A linear model, a logarithmic variant (assuming a log-normal distribution of the predictant, modelling the logarithm of operation density) and a Poisson model (typically used to model count variables). To provide robust results and prevent overfitting of the data, an appropriate subset of variables need

to be chosen from the set of available predictors. This is particularly important since some predictors are highly correlated amongst each other which is referred to as multicollinearity (Belsley et al. 1980). Even though multicollinearity does not reduce the predictive power of the model it may strongly affect the interpretation of individual regression coefficients of predictors containing mutual information. Besides being the cause of overfitting, it is thus desirable to reduce the number of (correlated) predictors to also better interpret resulting regression coefficients. To do so we chose an iterative procedure which

–starting from an initial model- stepwise removes or adds predictor variables. Which predictor to add to (or remove from) the list of predictors in the model is decided in each iterative step by maximization of the Akaike information criterion (AIC, compare Akaike, 1985). The basic idea is to assume a certain penalty for each (additional) predictor within the model. This penalty needs to be balanced to the resulting goodness of the model fit (e.g. by means of $R^2$) leading to an optimization problem between the total penalty and fit quality. The algorithm converges if no predictor can be added or removed to further optimise

the model in terms of the AIC. To perform this optimization procedure, the weight of the penalty can be varied by means of the parameter k. While k=2 corresponds to the classical AIC criterion, higher k result in an increased penalty for additional predictors. Different choices of k will ultimately lead to different optimized models including more (less) predictor variables if k is lower (higher).

### 3.4 Model validation methodology

To assess the predictive skill of the optimized model a cross validation is set up. For this, the area of Berlin is divided into four sectors. Then the model -using the set of predictors identified by means of the iterative procedure described above- is fitted four times, each time using all grid points within three of the four sectors. Each model fit is then used for predicting the operation density for grid points within the fourth sector. In this way predictions are obtained for data that has not been used

for model fitting. Calculating the mean squared error of these model predictions in comparison to observed operation density values results in the cross-validation error, which is used as the criterion for predictive model skill. The optimal choice of k in the iterative optimization procedure described above is not known up front. Different choices of k lead to differing number of predictor variables. Thus, to find the optimal model for predictive purposes, we vary k and compare the predictive model skill of the resulting models. The optimal choice is found by maximising the predictive model skill i.e. minimising the cross-validation error.

## 4 Results

### 4.1 Comparison of fire brigade operations and building damage data

Daily operations counts in the period 2002-2011 for the whole of Berlin are correlated to daily building losses in Berlin. Correlations are calculated for total operation counts, as well as counts for operations associated with individual alert keywords, additionally stratified by season. The aim of this analysis is to identify in how far specific categories of fire brigade operations (i.e. tree related operations) can be related to the wind and hail impacts as described by the insured loss data. Particularly because impact data cannot be directly related (e.g. missing flood damages in the insured loss data) it is valuable to analyse the relationships to gain an understanding on the causing events for fire brigade operations which is not readily available. However, this means that the interpretation of correlation results is difficult, since wind, hail and precipitation may occur simultaneously for both winter storms and thunderstorms. It is not directly clear if a certain correlation means that both datasets contain impacts due to the same meteorological factor (i.e. wind) or if correlations are due to the simultaneous occurrence of multiple meteorological factors.

Highest correlations are found between tree related operations and building damages, particularly in winter (0.74). In addition, operations associated to the alert keyword "construction element" show rather high correlations to building damages (0.67). In both cases, winter correlations are higher which indicates that a large share of these operations are caused by severe wind gusts. Counts of water damage operations in summer do not show any correlation to building losses, which is due to the fact, that flooding damages are not contained in the insurance data set available. In winter however, considerable correlation is found (0.41). It can be assumed that this correlation is because water related operations in winter often occur in conjunction with large-scale storm events which would indicate that precipitation impacts coincide with wind impacts on housing. Correlating tree-related with water-related operations gives further weight to this assumption. While correlation is considerable in winter (0.25), there is low correlation in summer (0.08). Similar results are found correlating operations related to the keyword "construction elements" and water-related operations. Thus, operations caused by severe winds (tree-related and construction elements) and water-related operations seem to occur mostly independently in summer, while in winter they seem to coincide more often. However, the low correlation between summer damages and water-related operations is still surprising. The fact that flooding damages to housing are not included in the loss dataset obviously leads to a non-existing correlation, if regarding only effects due to rainfall. Because thunderstorm events are often related to severe precipitation and in some case

to hail would suggest a certain correlation between hail-induced building damages and water-related operations in summer. The fact that no correlation is found, might in turn indicate that either hailfall is sufficiently rare to make up for a significant effect or, that hailfall impacts do not play a major role for the occurrence of operations.

Spatial patterns of insured losses and operation occurrences were compared for a set of 4 wind storm events (Kyrill, Emma, Xynthia and Lothar07) and 2 convectively driven summer events (Aram and Gunnar). A visual comparison of impacts for the winter storm Kyrill (2007/01/17-2007/01/19) and the thunderstorms related to the frontal passage of Gunnar (2011/06/22) can be found in Figure 1. In general, a rather good agreement in the patterns of the number of operations per zip code area and the number of insurance claims is found. For Kyrill, both datasets show considerably higher impacts in the south of Berlin, while
central and some northern parts of Berlin featured lower impacts. It can be argued that there is an influence of the size of areas that is not homogeneous (and particularly large areas are found in the south, while particularly small areas in central Berlin). However, considering relative numbers (normalizing for the zip code area) did not alter the qualitative findings. For the thunderstorms related to the frontal passage of Gunnar, spatial patterns also show considerable agreement. Affected areas are considerably larger when considering fire brigade operations, while building damages are more concentrated on individual zip
code areas. This might be related to hailfall that might have occurred localised leading to localised occurrence of building damages, while precipitation and wind gusts occurred more widespread leading to water-related operations and wind induced treefall in larger areas. A spatial correlation analysis is performed, correlating the number of insurance claims and the number of operations within each zip code area. It has been tested, that using different quantities (e.g. damage ratio and normalized operation densities) does not qualitatively influence the correlations. Also, it needs to be kept in mind that these spatial
correlations are evaluated only for individual events, which may thus not be generalized. Resulting spatial correlations for the 6 events are given in Table 2.

Most prominently, significant correlations are found for tree-related operations in relation to building damages. This might affirm that tree-related operations mostly represent wind-induced treefall, which relates directly to wind-induced building
damages. For some events (Kyrill, Lothar07 and Aram), considerable correlation is found for water related operations while for the others there is no correlation at all. While no direct water induced damages are included in loss dataset, there might be an indirect relation. For a specific event, severe precipitation might coincide with hailfall which themselves induce damages. For Lothar07 and Aram there are confirmed hail observations in Berlin or its surrounding. For Kyrill a study indicates that there has been thunderstorm activity during the frontal passage, which might have been related to hailfall (Fink et al., 2009).
The authors also note, that the severe precipitation could have increased damages. This might in turn explain why for Kyrill, Lothar07 and Aram, correlations for tree-related operations and building damages are particularly high.

Thus, it shows that the relationship is far from being an identity between building damages and fire brigade operations. Spatial patterns can be found to show considerable agreement in some cases, however individual impacts (i.e. different categories of

operations) are generated by multiple meteorological variables (severe gusts, precipitation and hail) or even by a complex interplay of those variables. Additional factors might distort the relationship between insured damages and operations. These include the fact that in case of major events both insurers as well as emergency services might alter their usual procedural strategies. For instance, insurances relinquish detailed plausibility checks for individual damage reports in case of cumulative

loss events. Also, emergency services request the public to handle non-life-threatening damages by themselves in certain situations to relieve workload for first responders. Both reasons might have contributed to the fact that for Kyrill an extremely high insured loss has been recorded (about 10 times higher compared to Lothar07) while the number of fire brigade operations is not as exceptional (comparable to Lothar07).

## 4.2 Spatial correlation between potential vulnerability predictors and patterns of operation occurrences

Patterns of average operation densities (represented by the number of operations per km² and per year) are calculated on a 1x1km grid (Figure 2). Considering all operations (Figure 2a), distinct spatial variations can be observed. In general, high densities are found in central areas of Berlin, while outskirts feature low densities. However, numerous additional spatial variations can be found, such as particularly low operation densities in less densely (or unsettled) areas such as the "Grunewald" and areas in the south-east of Berlin. But also for central parts of Berlin, distinct local minima in operation

densities are found, e.g. for the "Tierpark" or the former airport "Tempelhof". Considering individual alert keywords shows that patterns of the spatial densities of operations considerably vary. While water-related operations show a rather similar spatial pattern compared with all operations, operations related to traffic-obstructions or treefall are distributed rather differently. Both are distributed more broadly over the area of Berlin, not featuring the distinct concentration on the centre. Furthermore, for operations related to traffic obstructions a concentration of emergency operations near important junctions

can be found (Figure 2c). For tree related operations, it seems that maxima of operation occurrence are not found in forest areas themselves but rather at their borders with housing areas (e.g. compare the border areas of the "Grunewald" in Figure 2d). This is not unexpected, since major impacts due to treefall is not expected in wooden areas but particularly in areas where trees are present in the direct vicinity of man-made structures (e.g. roadside trees or trees in recreational areas). This implies that only in very few cases the modelling of vulnerabilities to (meteorological) hazards can be made in a univariate fashion.

Instead, combinations of multiple factors will determine local vulnerability and consequently those need to be considered.

Examples for the spatial patterns of potential predictors for vulnerability are given in Figure 3. Even though building density (shown in Figure 3a) and building coverage (Figure 3b) are based on the same data (i.e. individual housing information as derived from open street map), different information can be extracted. While building density is calculated as the number of

houses per square kilometre, building coverage assesses the area fraction covered by buildings. Hence, building density is particularly high in suburban areas with numerous small houses while building coverage is highest in central areas with concentrated large buildings. Similarly, information on the density of the road network can be derived from Open Street Map (Figure 3c). Additional predictor variables from the CORINE land cover dataset are assessed, by calculating the fraction of a

grid box that is covered by areas of a specific CORINE land use type (as one example, Figure 3d shows the area fraction of artificial surfaces). Again, quite different information can be gathered, e.g. when considering the different land use types encoded in CORINE. Finally, with respect to the aim of modelling local vulnerabilities, the characteristics of the local urban structure can be described on the basis of not only one but instead multiple of these predictor variables.

For the predictor variables listed in Table 3, the spatial correlation to the gridded operations densities is calculated. For this, only grid points within Berlin are considered for which data on operations is available. Furthermore, correlation is assessed for individual alert keywords, as well as considering all operations. Resulting correlations are listed in Table 3, with colours indicating positive correlation (in red) and negative correlations (in blue). Several predictor variables stand out in this table, in particular the building coverage and the area fraction of continuous urban fabric, which have high correlations with spatial patterns of operations disregarding their alert keyword. One exception are tree-related operations for which correlation to both building coverage and area fraction of continuous urban fabric are considerably lower. Instead, in this case correlations are rather high for building density and the area fraction of discontinuous urban fabric. It might be assumed that this is due to the fact, that particularly in the outskirts of Berlin (with a high number of small buildings) the vulnerability is increased due to the presence of trees in gardens (i.e. in vicinity to buildings). However, in general it can be deduced that the degree of urbanisation (both expressed by the area coverage of housing and indicated by continuous urban fabric areas) plays a major role in determining highly vulnerable areas. Both variables can be interpreted as a proxy for the number of "objects" at harm (e.g. the number of basements or drainage systems in case of water related emergencies). For tree-related operations, the picture is quite different however. Operation densities are particularly high in areas of discontinuous urban fabric and seem to be enhanced in areas of high building densities (i.e. the number of houses per km²). Both indicates, that tree-related operations are more likely in less densely covered urban areas, where assumedly more roadside trees or trees as part of recreational areas can be found in close vicinity to building structures. Considering area fraction of wooded areas (particularly coniferous forests), negative correlations with operations of all alert keywords are found. This can be explained by the fact, that this variable is essentially inverted to areas with a high fraction covered by urban structures. Interestingly, also tree-related operations are negatively correlated to areas with a high fraction of wooded areas. This indicated that it is not areas with many trees which are particularly vulnerable, but instead areas in which trees are found in vicinity of man-made structures. Considering the density of the road network it is found that positive correlation to the patterns of each individual alert keyword exist. Particularly this holds for the secondary and tertiary road network. A simple explanation for this is that areas in which a high density of secondary and tertiary roads exist mostly coincide with areas of high building coverage. Additionally, it can be found, that correlations of road density patterns are highest with respect to operations related to traffic obstruction. Obviously, this is due to the fact that traffic obstructions are more likely to occur in areas with a high density of roads. All the above-mentioned findings show, that even though there is no complete correspondence between individual predictors and the occurrence of operations, numerous predictors can be found explaining a share of the spatial variability of weather impacts.

This shall be exploited in the following, by building multivariate models to statistically describe the spatial patterns of operation occurrences.

**4.3 Multi-variate modelling of the occurrence of fire brigade operations**

The set of predictors described above is used to set up a multi-variate model to be able to predict the local occurrence rate of operations. As described in Section 3, the iterative procedure consists of the repeated application of a parameter selection algorithm while iteratively increasing the penalty for additional model parameters. The optimal model is then chosen by means of the cross-validation error (to prevent overfitting and estimate the predictive ability of the resulting model). As an example, the results of this iterative procedure are shown in Figure 4 for water related operations. For the linear model the optimal model is identified having 12 predictor variables (which is the result of choosing k=1 as the penalty weight). Choosing a lower weight

of zero obviously results in a model taking into account all 33 possible predictors featuring severe overfitting (mscve is about 4000 in this case).  The procedure is applied to fire brigade operations associated to individual alert keywords, as well as all operations together. For the latter, the resulting optimal model includes a set of 12 predictor variables (listed in Table 4), explaining 83% of the variance in the spatial pattern of operations. In accordance to the correlation analysis, the predictor

"building coverage" possesses the highest contribution to the explained variance (EV=59%) while for other variables lower contributions are found (e.g. "area fraction continuous urban fabric" contributes 11% and "building density" 6%). Of course, there is not a direct correspondence when comparing the EV of individual predictor variables to the correlations as listed in Table 4, since certain predictor variables might be strongly correlated (multicollinearity, compare Section 3.3). By adding a predictor which is correlated to predictors already in the model, increase in model performance might be small even though

the correlation to the predictant is high. The results described above apply to a basic linear model. Alternatively, the predictor selection methodology can be applied while using alternative models, i.e. a log-normal and a Poisson model. Results showed that in general predictive abilities of the statistical models (in terms of the cross-validation error) are not increased (not shown). By means of the mean squared cross validation error (msvce), the linear models appear to perform best. However, the linear model suffers from the disadvantage of predicting negative values for the number of emergency calls in some cases, while

both log-normal and Poisson model do not. It should be noted that the different models may contain a different set and even different number of predictors. The comparison of individual regression parameters is thus difficult. However, models can be compared in terms of their predictive skill. In the following, results are shown using the linear model, performing best in terms of the predictive skill (assessed by means of the mscve).

In comparison to the maps shown in Figure 2, Figure 5 shows model predictions for the average number of operations on a 1x1km grid cell. In the case considering all emergency operations (Figure 5a) or only water related (Figure 5b), the model nicely reproduces the concentration of operation occurrences in central parts of Berlin, while especially forest areas such as the Grunewald feature very low occurrence rates. In addition, the amplitude of this variation (ranging from 0 to about 80

operations per km² per year considering all operations) is well captured. Individual hotspots of high operation occurrence rates however are only partly reproduced. Particularly this is the case for a hotspot in the south-east of Berlin centre (Figure 2a and 2b, corresponding to northern parts of the district "Neukölln"). It can be found that in this area particularly water related operations are very high. It is possible that this is influenced by an extraordinarily high population density in these areas, an

information which is only partly (and indirectly) covered by predictor variables such as building coverage. Also, other factors such as housing conditions or very localized troughs (potentially leading to water accumulation in case of severe rain) might of course affect the occurrence of emergency operations. Such information however has not been available for this study and thus cannot be taken into account.

Also in case of traffic- and tree- related operations, predicted patterns (shown in Figure 5 c and d) reproduce observed patterns

rather well. In both cases occurrence rates are less concentrated on central parts of Berlin but are more widely distributed across Berlin. Particularly in case of tree related operations, model predictions show a rather homogeneous distribution over large parts of Berlin (Figure 5d), while local maxima in the observed operation density (Figure 2d) are poorly captured. Considering the explained variation (EV) for the different models, it is confirmed that for tree-related operations the predictive ability of the model is worst, with an EV of 53%. In comparison, the model for all operations has an explained variation of

83% (Table 4).

## 5 Conclusions and Discussion

A comparison of a new data set containing spatial and temporal information on emergency operations of the Berlin fire brigade with damage data has been performed. Spatial patterns can be derived and correspondences amongst both impact data sets can be found. However, a complex interplay of meteorological conditions leads to a variety of weather impacts, making it very

hard to directly compare the datasets. Instead, the availability of both datasets might be considered as particularly valuable for the reconstructing the multifaceted impacts of severe weather events.

The relation to predictor variables for the structure of settlement as well as characteristics on land use has been addressed by means of an analysis of spatial correlations. Particularly the information on the local building coverage and shows a rather high influence on the occurrence of operations. Accordingly, areas classified as continuous urban fabric (within the CORINE

land cover dataset) exhibit high rates of fire brigade operations. Analysing individual alert keywords, other variables turn out as valuable predictors. E.g. in case of traffic related operation these include the local density of the road network. In case of tree related operations particularly the areas classified as discontinuous urban fabric correlate with high occurrence rates. An interpretation for this is that in these areas a higher number of trees are present in the direct vicinity of man-made structures (e.g. roadside trees or trees in recreational areas).

Multi-variate modelling including an iterative prediction selection algorithm has been conducted, with resulting models being able to predict the local vulnerabilities. Evaluation of models showed moderate model performances for tree-related operation occurrences (explained variance of 53%), while for other types of operations – i.e. water related, traffic related, or all operations combined- model results were better (explained variance of 70 - 80%). In all cases, spatial patterns of operation occurrences

can be reproduced well. Except for tree related operations, the amplitude of variations can also be reproduced. However, individual hotspots with high occurrence rates are only insufficiently predicted indicating that particular information influencing the local vulnerabilities are not included in the predictor variables available in this study. In case of water related operations, these might for example include housing conditions. Also, information on local tree stocks, particularly in vicinity of vulnerable structures might be very valuable to better model tree-related operation occurrences.

As one important result which can be the starting point for further investigations, Table 4 can serve as an overview of the most relevant parameters to describe the local vulnerability to severe weather. While many details on individual predictor variables and their descriptive power with respect to specific types of operations can be found, a common feature is identified, namely the fact that the building coverage is by far the most dominant factor to describe the local vulnerability.

The model has been developed and tested for the Berlin area, due to the availability of fire brigade operation records for Berlin. Yet, model predictions can be derived for the whole of Germany. Such model predictions might be particularly valuable for regions with no systematic records on weather impacts. However, such extrapolation might suffer from potentially severe limitations. The occurrence of severe weather conditions is not homogenous over Germany, with storm frequencies being higher in northern regions and thunderstorm frequencies being higher in southern regions. Thus, the distribution of hazards causing local impacts can differ considerably, which will certainly affect the occurrence of emergency operations. Such effects are excluded in the presented modelling approach, which assume a homogenous distribution of hazards. For the investigation are of Berlin this is certainly a valid assumption. Extracting model predictions for other urban areas might suffer from an offset in terms of absolute number of operations. Such model predictions can however still be very valuable since they can provide information on spatial variation in operation occurrences on a sub-city scale. Still, future work should include also meteorological/climatological information on different hazards, which will strongly influence local vulnerability and thus predicted weather impacts.

The presented model to predict the local vulnerability to severe weather can serve as a basis for a broad range of tools or applications in emergency management. These might include tools for the long-term resource planning of local emergency management capacities. Also, handling short-term variations in the demand of local emergency management capacities might be supported by such tools when including actual weather information. In this study, we focussed on datasets which are publicly available -partly open source community data- which are all available area-wide for at least the whole of Europe. This yields great potential for the design of national or even pan-European tools and applications in emergency management.

**Acknowledgement**

This research was carried out in the Hans-Ertel-Centre for Weather Research (Simmer et al., 2016). This research network of Universities, Research Institutes and the Deutscher Wetterdienst is funded by the BMVI (Federal Ministry of Transport and

Digital Infrastructures). We wish to thank the Gesamtverband der Deutschen Versicherungswirtschaft e.V. (GDV) for providing the loss data and for many useful discussions. We thank the Berlin fire brigade to provide the data set on weather related operations. Finally, we thank O. Petrucci and one anonymous reviewer for their helpful comments as well as V. Kotroni for editing our manuscript and therefore helped to improve this manuscript.

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

**Tables**

| | Full year | | | Summer | | Winter | |
|---|---|---|---|---|---|---|---|
| | Absolute number | Total fraction | correlation with daily losses | Summer share | correlation with daily losses | Winter share | correlation with daily losses |
| All Operations | 10.069 | 100% | 0.58*** | 50 % | 0.59*** | 50 % | 0.57*** |
| Water damage | 3.358 | 33 % | 0.14*** | 53 % | 0.06 | 47 % | 0.41*** |
| Traffic obstruction | 2.549 | 25 % | 0.22*** | 46 % | 0.15*** | 54 % | 0.30*** |
| Tree related | 1.715 | 17 % | 0.67*** | 73 % | 0.69*** | 27 % | 0.74*** |
| Construction element | 1.407 | 14 % | 0.58*** | 45 % | 0.59*** | 55 % | 0.67*** |
| Ice and snow | 211 | 2 % | 0.00 | 0 % | - | 100 % | 0.00 |
| Others | 831 | 8 % | 0.00 | 18 % | 0.01 | 82 % | 0.01 |

**Table 1: Distributions of impacts of different types in Berlin for the period 2002-2011, stratified by their cause and by season, i.e.**

5   **winter and summer half year and temporal correlations to daily building damages. Significance is indicated if p-value is below 0.05 (\*), below 0.01 (\*\*) and below 0.001 (\*\*\*).**

| | **Kyrill** | **Emma** | **Xynthia** | **Lothar07** | **Aram+** | **Gunnar+** |
|---|---|---|---|---|---|---|
| All Operations | 0.70*** | 0.20** | 0.09 | 0.78*** | 0.57*** | 0.46*** |
| Water damages | 0.45*** | -0.05 | -0.06 | 0.54*** | 0.20** | 0.06 |
| Traffic obstruction | 0.09 | -0.08 | 0.11 | 0.45*** | 0.16* | -0.04 |
| Tree related | 0.73*** | 0.42*** | 0.28*** | 0.79*** | 0.79*** | 0.58*** |
| Construction element | 0.03 | -0.04 | -0.1 | 0.21** | 0.19* | -0.07 |
| Others | 0.06 | -0.05 | -0.01 | -0.06 | 0.17* | 0.01 |

**Table 2: Spatial correlations for specific events. The correlation is calculated between the number of fire brigade operations and the number of insurance claims within individual zip code areas. Significance is indicated if p-value is below 0.05 (*), below 0.01 (**) and below 0.001 (***).**

| Predictor | All | Water related | Traffic obstruct. | Tree related | Constr. element | Ice & snow | Other |
|---|---|---|---|---|---|---|---|
| Building Density | 0,24*** | 0,21*** | 0,16*** | 0,49*** | 0,18*** | 0,15*** | 0,17*** |
| Building Cover | 0,79*** | 0,71*** | 0,72*** | 0,62*** | 0,73*** | 0,64*** | 0,7*** |
| Street Density (All) | 0,48*** | 0,38*** | 0,58*** | 0,43*** | 0,42*** | 0,29*** | 0,38*** |
| Street Density (Motorway) | 0,05 | 0,02 | 0,14*** | 0,03 | 0,03 | -0,01 | 0,01 |
| Street Density (Primary ) | 0,36*** | 0,27*** | 0,53*** | 0,18*** | 0,32*** | 0,23*** | 0,29*** |
| Street Density (Secondary ) | 0,54*** | 0,45*** | 0,59*** | 0,39*** | 0,5*** | 0,43*** | 0,47*** |
| Street Density (Tertiary) | 0,34*** | 0,29*** | 0,37*** | 0,34*** | 0,3*** | 0,21*** | 0,28*** |
| Street Density (Other) | 0,37*** | 0,29*** | 0,45*** | 0,36*** | 0,31*** | 0,21*** | 0,28*** |
| Orographic Height | -0,11** | -0,1** | -0,06 | -0,01 | -0,14*** | -0,15*** | -0,14*** |
| Orographic Slope | 0,02 | 0,02 | -0,02 | 0,02 | 0,03 | 0,04 | 0,03 |
| AF Continuous urban fabric | 0,77*** | 0,71*** | 0,61*** | 0,31*** | 0,81*** | 0,79*** | 0,81*** |
| AF Discontinuous urban fabric | 0,25*** | 0,2*** | 0,29*** | 0,58*** | 0,13*** | 0,04 | 0,1** |
| AF Industrial or commercial units | -0,03 | -0,05 | 0,1** | -0,11** | -0,03 | -0,05 | -0,06 |
| AF Industrial or commercial units | 0,04 | 0,03 | 0,08* | 0,03 | 0,04 | 0,02 | 0,03 |
| AF Port areas | 0,01 | -0,01 | 0,08* | 0 | -0,01 | -0,01 | -0,01 |
| AF Airports | -0,03 | -0,02 | -0,03 | -0,09** | -0,01 | 0 | -0,01 |
| AF Dump sites | -0,02 | -0,02 | -0,03 | -0,03 | -0,02 | -0,01 | -0,02 |
| AF Construction sites | 0,06 | 0,05 | 0,07* | 0,03 | 0,05 | 0,04 | 0,08* |
| AF Green urban areas | 0 | -0,01 | 0,02 | -0,04 | 0 | -0,01 | -0,01 |
| AF Sport and leisure facilities | -0,1** | -0,1** | -0,06 | -0,08* | -0,1** | -0,09** | -0,09** |
| AF Non-irrigated arable land | -0,18*** | -0,15*** | -0,2*** | -0,21*** | -0,15*** | -0,11** | -0,13*** |
| AF Fruit trees & berry plantations | -0,02 | -0,02 | -0,02 | -0,02 | -0,01 | -0,02 | -0,02 |
| AF Pastures | -0,09** | -0,08* | -0,1** | -0,1** | -0,08* | -0,06 | -0,07* |
| AF Complex cultivation patterns | -0,04 | -0,03 | -0,04 | -0,05 | -0,03 | -0,02 | -0,03 |
| AF Agricultural land | -0,08* | -0,06 | -0,09* | -0,11*** | -0,06 | -0,05 | -0,06 |
| AF Broad-leaved forest | -0,22*** | -0,19*** | -0,25*** | -0,21*** | -0,19*** | -0,14*** | -0,17*** |
| AF Coniferous forest | -0,29*** | -0,24*** | -0,33*** | -0,37*** | -0,23*** | -0,17*** | -0,21*** |
| AF Mixed forest | -0,15*** | -0,12*** | -0,18*** | -0,16*** | -0,13*** | -0,09** | -0,11*** |
| AF Natural grasslands | -0,05 | -0,04 | -0,04 | -0,07* | -0,04 | -0,03 | -0,03 |
| AF Transitional woodland-shrub | -0,08* | -0,06 | -0,08* | -0,11** | -0,06 | -0,04 | -0,05 |
| AF Inland marshes | -0,05 | -0,04 | -0,05 | -0,05 | -0,04 | -0,03 | -0,03 |
| AF Water courses | 0,01 | -0,01 | 0,06 | -0,04 | 0,01 | 0,01 | 0,01 |
| AF Water bodies | -0,17*** | -0,14*** | -0,2*** | -0,2*** | -0,14*** | -0,09** | -0,12*** |

**Table 3: Spatial correlation coefficients (Pearson correlation) between yearly averaged operation density with exposure predictors. Some CORINE Classes are excluded in this table, if there are no areas in Berlin, hence the area fraction (AF) is 0 everywhere. High/low correlations are highlighted in red/blue. Significance is indicated if p-value is below 0.05 (*), below 0.01 (**) and below 0.001 (***).**

| Model | Predictor | Estimate | Std. Error | Significance | EV [%] |
|---|---|---|---|---|---|
| **All Operations** | Building Cover | 12.7 | 6.5 | *** | 58.8 |
| 12 predictors; EV: 83% | Area Fraction "Continuous urban fabric" | 31.6 | 3.0 | *** | 10.8 |
| | Building Density | -0.0031 | 0.0009 | *** | 5.9 |
| | Area Fraction "Industrial or commercial units" | -22.8 | 1.9 | *** | 2.8 |
| | Street Density (Secondary) | 2.01 | 0.26 | *** | 2.6 |
| | Street Density (Primary) | 1.72 | 0.34 | *** | 1.2 |
| **Water related** | Building Cover | 62.3 | 3.8 | *** | 47.8 |
| 7 predictors, EV: 69% | Area Fraction "Industrial or commercial units" | -11.3 | 1.1 | *** | 11.3 |
| | Building density | -0.0017 | 0.0005 | *** | 4.2 |
| | Area Fraction "Continuous urban fabric" | 12.2 | 1.8 | *** | 1.7 |
| | Area Fraction "Discontinuous urban fabric" | -11.3 | 1.1 | *** | 1.5 |
| **Traffic obstruction** | Building Cover | 14.1 | 1.3 | *** | 53.9 |
| 8 predictors, EV: 78% | Street Density (Primary) | 1.65 | 0.08 | *** | 9.3 |
| | Street Density (Secondary) | 1.03 | 0.06 | *** | 7.7 |
| | Building Density | -0.0010 | 0.0002 | *** | 2.6 |
| | Area Fraction "Continuous urban fabric" | 4.0 | 0.5 | *** | 2.1 |
| | Street Density (Motorway) | 1.0 | 0.1 | *** | 1.5 |
| **Tree related** | Building Cover | 11.4 | 0.6 | *** | 38.7 |
| 4 predictors, EV: 53% | Area Fraction "Discontinuous urban fabric" | 1.4 | 0.1 | *** | 9.5 |
| | Area Fraction "Industrial or commercial units" | -2.4 | 0.3 | *** | 3.8 |
| **Construction element** | Building Cover | 23.2 | 1.2 | *** | 54.3 |
| 8 predictors, EV: 81% | Area Fraction "Industrial or commercial units" | -4.2 | 0.4 | *** | 12.9 |
| | Area Fraction "Discontinuous urban fabric" | -1.6 | 0.2 | *** | 5.3 |
| | Area Fraction "Continuous urban fabric" | 7.8 | 0.6 | *** | 4.0 |
| | Building Density | -0.0005 | 0.0001 | ** | 3.2 |
| **Ice and snow** | Building Cover | 4.5 | 0.3 | *** | 40.4 |
| 8 predictors, EV: 72% | Area Fraction "Continuous urban fabric" | 1.9 | 0.2 | *** | 24.7 |
| | Street Density (Secondary) | 0.05 | 0.01 | *** | 2.7 |
| | Area Fraction "Industrial or commercial units" | -0.9 | 0.1 | *** | 2.3 |
| | Street Density (All) | -0.006 | 0.002 | * | 1.6 |
| **Others** | Building Cover | 14.8 | 0.9 | *** | 48.5 |
| 6 predictors, EV: 78% | Area Fraction "Industrial or commercial units" | -3.0 | 0.3 | *** | 14.2 |
| | Area Fraction "Discontinuous urban fabric" | -1.4 | 0.2 | *** | 9.0 |
| | Area Fraction "Continuous urban fabric" | 5.3 | 0.4 | *** | 4.1 |
| | Street Density (Secondary) | 0.14 | 0.03 | *** | 1.1 |
| | Area Fraction "Sport and leisure facilities" | -1.1 | 0.3 | *** | 1.1 |

**Table 4: Resulting optimal models. First column indicates the number of predictors as well as the total explained variation (EV) of the chosen optimal model (according to the cross validation error). The leading predictors of each model are shown indicating weather a positive (+) or negative effect (-) is found. In the last column, explained variance in % is given for these leading predictors. Within the table, predictors are shown if they have an explained variance > 1 %. Significance is indicated if p-value is below 0.05 (*), below 0.01 (**) and below 0.001 (***).**

**Figures**

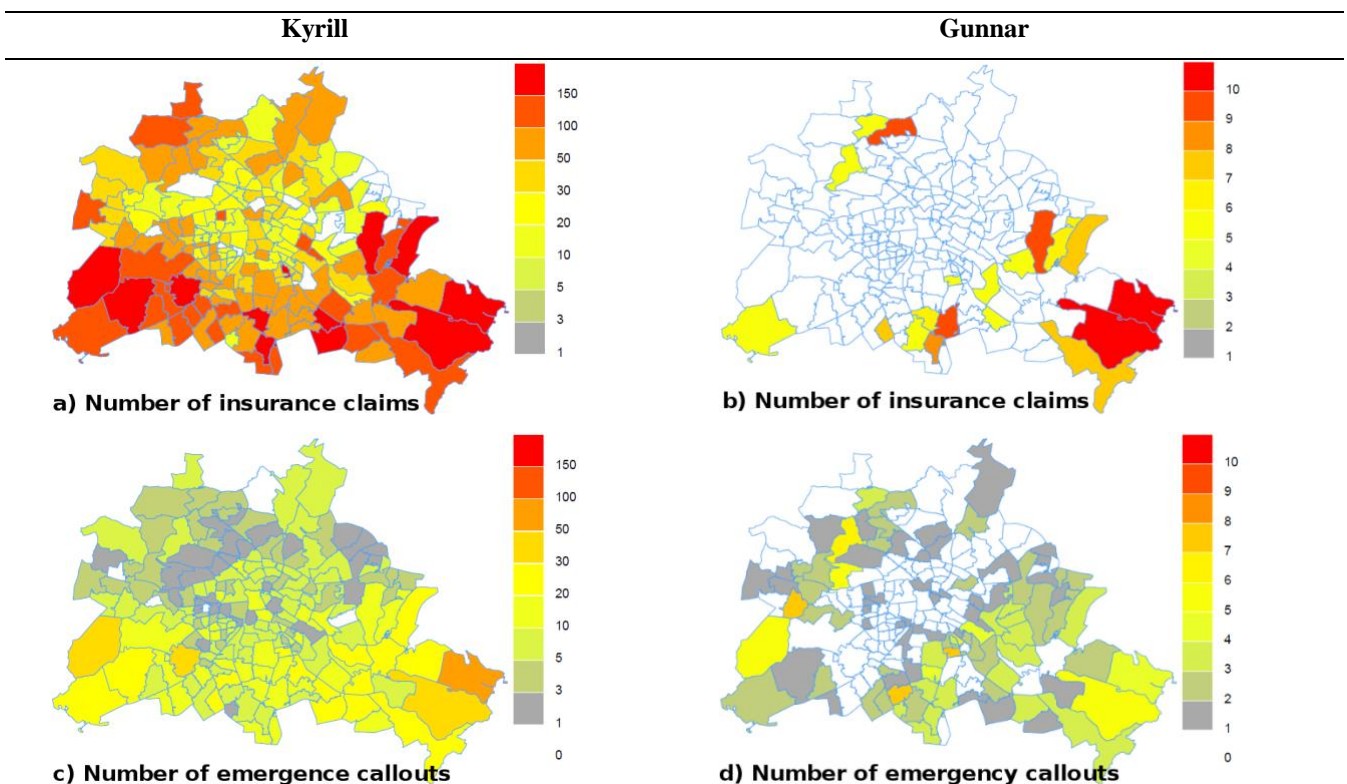

**Figure 1: Spatial comparison of the number of insurance claims (top row) to the number of fire brigade operations (bottom row) for winter storm "Kyrill" in 2007 (left) and frontal passage "Gunnar" in 2011 (right).**

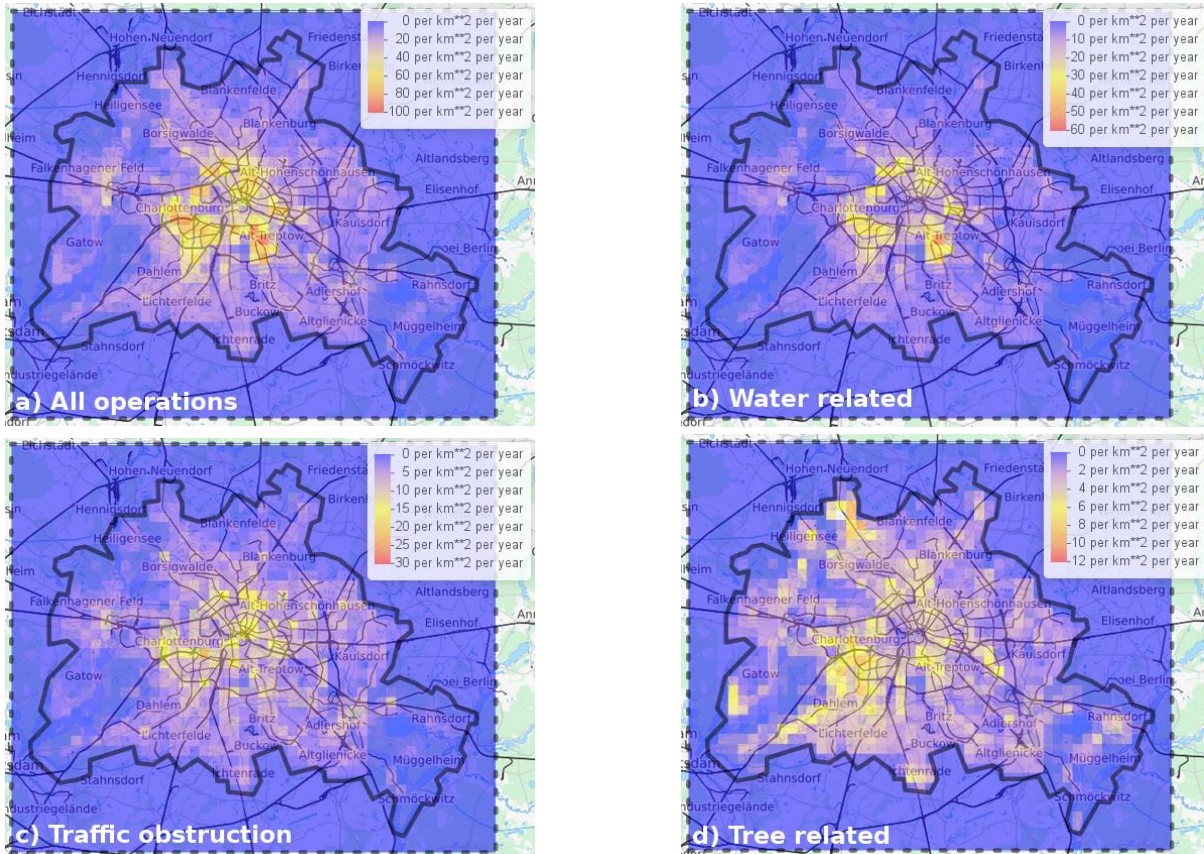

**Figure 2: Mean yearly density of fire brigade operations during 2002-2011 calculated on a 1x1km grid [units: operations per km² per year]. Operation recordings are available for Berlin only (boundaries are indicated by black solid lines), i.e. zero values outside of Berlin are due to unavailability of data. Note the different colouring scale due to the fact that the absolute numbers of operations for a certain operation type vary considerably.**

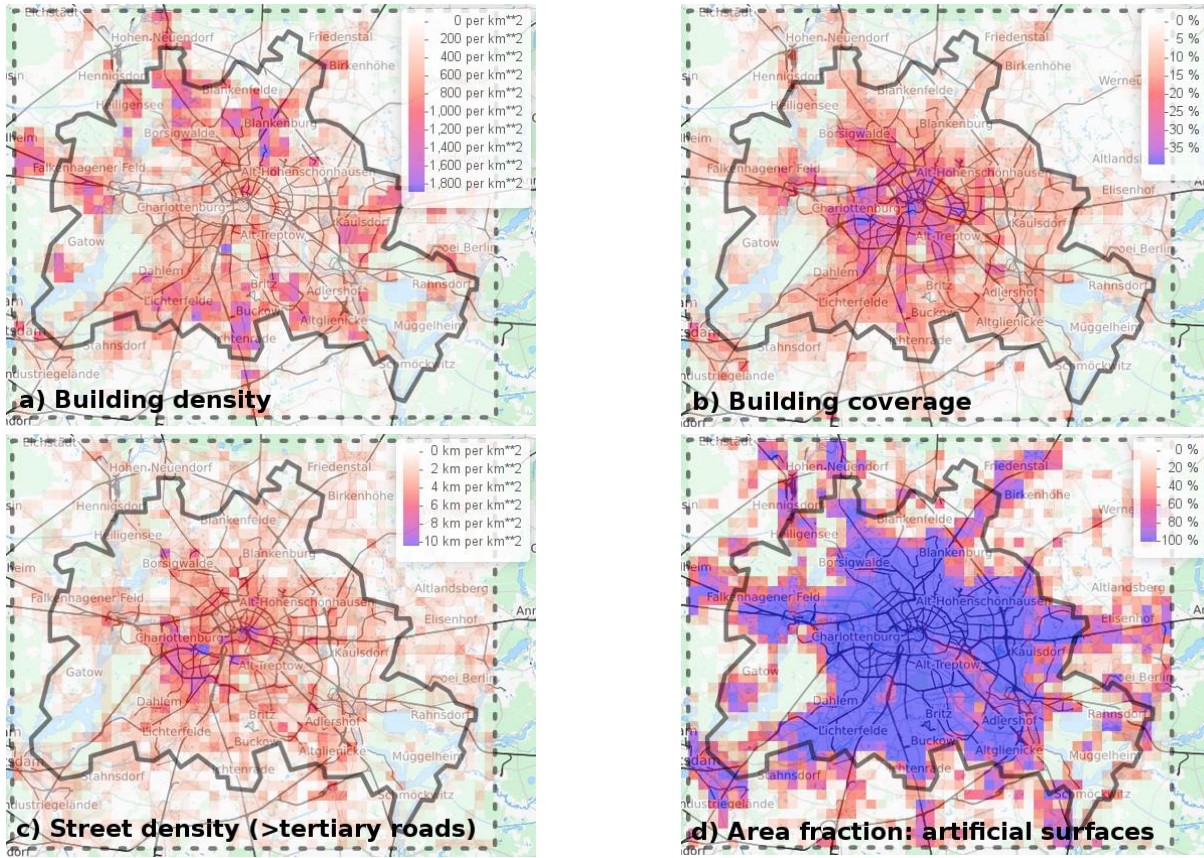

**Figure 3: Example set of exposure predictors calculated on a 1 x 1km grid. While building density (a), building coverage (b) and street density for tertiary roads and higher (c) are based on information extracted from open street map data, (d) shows the area fraction of artificial surfaces as derived from the CORINE land cover dataset.**

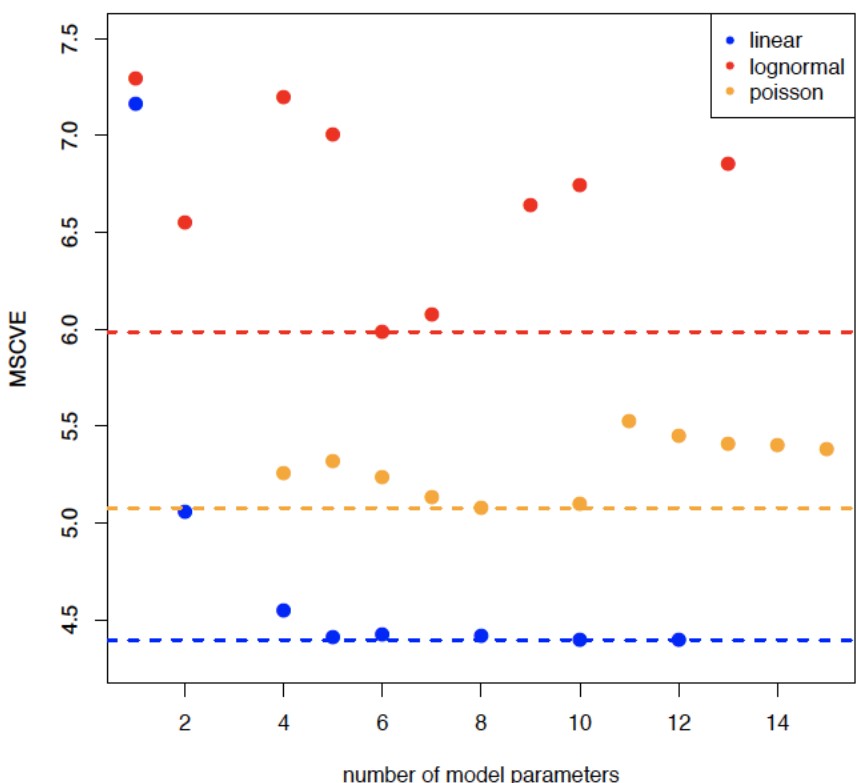

**Figure 4: Results of the iterative procedure to optimize the regression model. Increasing the penalty term for additional predictors leads to a model with smaller sets of predictor variables. For each of the resulting predictor sets and the corresponding multiple regression, the mean cross validation error (MSCVE) is calculated and plotted here. Blue circles represent validation results using a linear regression, orange circles represent results using the lognormal model and red circles represents results using a poisson regression.**

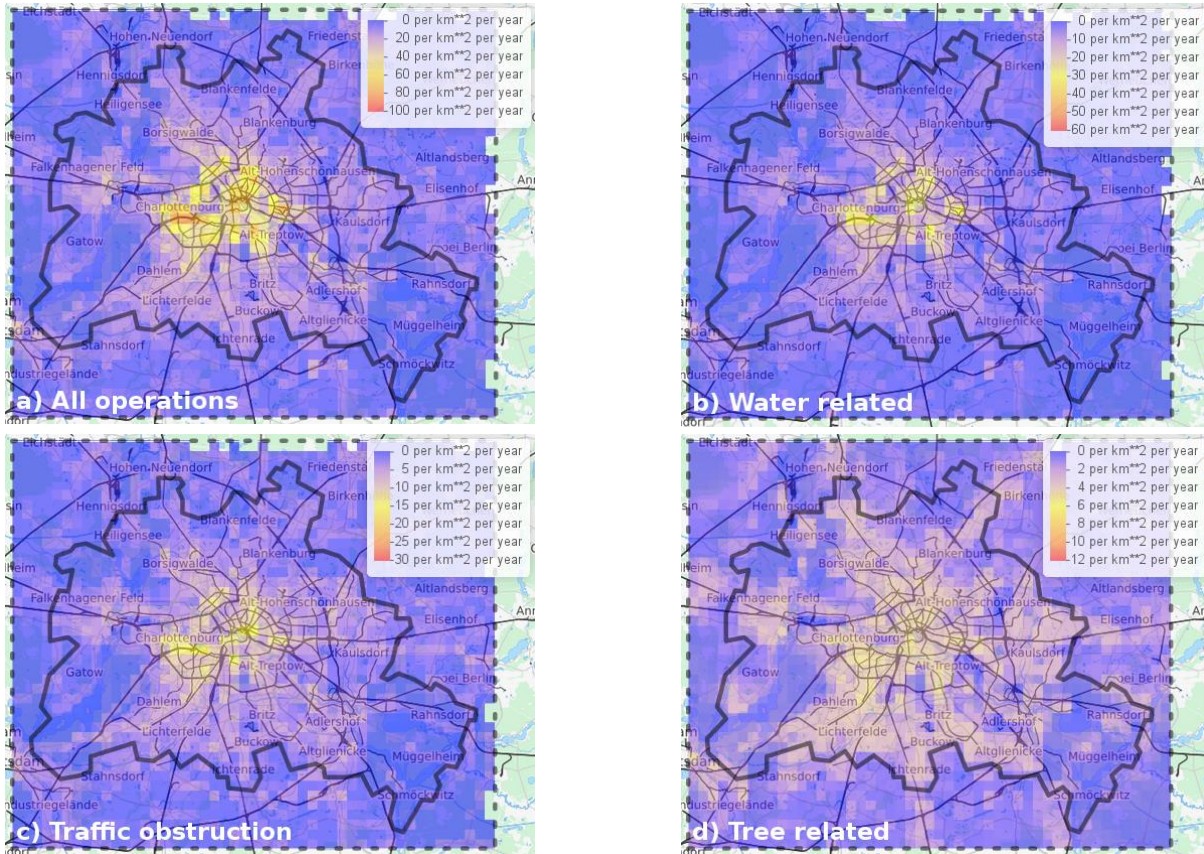

**Figure 5: Modelled mean yearly density of fire brigade operations [units: operations per km² per year]. (a) Results are shown for the model including all operations disregarding their type (a), for water related operations (b), for traffic obstructions (c) and for tree related operations (d).**