# Peer review of "A Statistical Model to Estimate the Local Vulnerability to Severe Weather"

_Natural Hazards and Earth System Sciences, 2017_

## Referee Comment (RC1) · Anonymous Referee #1 · 11 Oct 2017

The concept of the work is interesting; however there are several problems related to the methodology, the presentation of the results, the contribution of the resulting model, and consistency within the text. Furthermore, special attention to English language has to be paid, as there are several sentences that need to be corrected or rephrased.

Major issues: 1) A major issue is the statistical methods applied and the presentation of statistical results. Nowhere in the text or Tables are any notes provided regarding the calculated correlation coefficient. What kind of correlation (pearson's r, spearman's rho...??), are data normally distributed, any tests made for that, what are the p-values for testing significance of results. P-values are actually very important, along with the relevant significance limits (e.g. <0.05, most commonly used). The same for regression results, for which tables must provide also the standard errors, whether the coefficients are standardized, and what is the overall significance of the model (this I assume is high judging from the R2). Also, the number of observations for each correlation and

regression must be provided.

2) Important methodological deficiency is related to:

a) Failing to address the exact weather cause of operations and losses. The operations alert keywords do not necessarily show the weather cause. This could be done only by using meteorological stations near the operations and include meteorological parameters in the analysis because these are the hazards.

b) Lack of flood-specific operation or insurance data. Floods constitute the most damaging and expensive weather-related data. I would expect this to be clarified. Floods can cause damage to buildings, cars, trees, displacements, thus the model would have a serious value. In addition, it is not clear whether loss data have been identified as to the weather-related cause. It seems that loss data are not distributed according to the cause(s), while only windstorms/thunderstorms are addressed.

3. The actual contribution of the models is pending, since the predictor factors chosen ('...topographic features, land use, urban structure) are more or less constant throughout the examined period, and depict only the structure of the city. Predictors would have been the meteorological hazard parameters, which are missing from the models. I suggest the authors identify the daily weather conditions, based on meteorological data from weather stations in the city. Otherwise, the models only depict what has happened in the past based on an unchangeable 'environment'. Prediction based only on stable (endogenous) factors is not a prediction. Even the season parameter, for which a discussion was made in the methodology, is not included in the models as a dummy variable. Out of the two objectives set by the authors in the introduction (p2, lines 24-26), only the second is to my opinion reached: the identification of hotspots for weather impacts. It is not possible to 'predict the local occurrence rate of operations (p. 10, I13)' based only on these vulnerability indicators and on regressions of annual values.

4. Following the previous comment, there is a high correlation between the predictor

factors and the outcomes. Specifically, I find it logical for the road network density to be highly correlated with traffic problems, or housing density with construction-related impacts. This may be a methodological problem for multiple regressions. However, I can't evaluate the regressions without the required additional statistical information.

5. The authors use the expression 'potential predictors'. Then, it must be noted exactly what is the methodological contribution, because a serious problem for scholars is to find suitable and reliable predictors for relevant studies. Again, the problem is that meteorological variables or other hazard-related variables are not included or discussed in the present analysis.

**3) Other major issues**

Section 2.1. Are flood-related damages included? Please specify, since floods are the costliest weather-related hazards.

Section 2.2. Berlin-wide damages are available on daily basis for the period 1997-2011, while data on zip code level (190 within Berlin) is available for selected events only: how many events? In general, this paragraph is not clear. If only wind and light-ning are included, how can the comparison with operations be made? The selection of 5 events was based on which criteria?

P.4. line 1: evaluated for what exactly? Relating to the spatial distribution of operations? And why in section 4.1 you mention 6 events instead of 5?

P.4. line 5: the second part of the sentence – regarding hail- is based on other references? If it is so, consider separating the sentence, because the first part refers to the analysis of losses, thus it is part of the work done by the authors.

Section 3.1. The comparison of operations and loss data is strange. As explained in section 2.2, loss data refer only to wind and thunderstorms, while operations may be related to many other causes. So, what is correlated? It is confusing.

Section 4.1. In general, there is an inconsistency in these correlations since flood-

**C3**

related losses are not included. Many assumptions have been made in the interpretation of results for this reason. E.g.: P7,I2 'coinciding with roofing damages or other wind-caused building damages' Is this an assumption? Since losses do not include flood-related damages, correlations for the water-related categories are not useful. Operations and losses are very likely to refer to different hazards.

Other methodological issues:

In the Abstract the building density is suggested to be the most influential predictor. However, results suggest that building coverage has the strongest effect.

The expression 'water-related' operation refers to what specifically? Other than flooding, or flooding is included?

I would suggest you explain in the text what 'operations are equally distributed over winter and summer half year' mean. E.g. in a parenthesis (e.g. May to October, for summer half). Also, some local weather info would be helpful. For example, during summer half-year is there more wind and storms? And why do you make this 2-seasons choice?

Why is population density not included in the predictors?

What about the validation results? Why they are not shown?

References in the introduction regarding the use of fire service operations as impact indicator are missing. However, there are relevant papers.

What does a correlation between predictors and the TOTAL number of operations show, since damages can be due to different/various causes?

Minor: Please, follow the same citation format (e.g. see differences in page 2, lines 5 and then 7.

Please, follow the same term for fire brigade operations. Are they callouts or operations? The authors use different terminology. However, callouts and final operations usually are 2 different things. In page 8, scetion4.2, the term used is 'operation densities'. However, it is still the number used.

P.4. line 2: is it storm or thunderstorm specifically?

Sections 3.2, 4.2. Consider improving the title of this section. Correlation of What (?) with potential vulnerability predictors.

P9, I12: 'Additional predictor variables from the CORINE land cover dataset are assessed'. Only one is shown in the discussed picture (artificial surfaces)

P10.I9: Consider removing 'considerable'. This is to be evaluated in the regression results.

Sentences that should be definitely corrected/rephrased (not all cases are included. The article needs to be carefully reviewed for grammar):

P2, lines 12-13.

P3, lines 14-15.

P4. Lines 1-2

P4. Lines 14-15 (consider changing the beginning of the sentence)

P7. Lines 22-23. A verb is missing.

P7, I6: 'Correlating tree-related and water-related operations'. Consider correcting as: Correlating tree-related WITH water-related operations?

P8, lines 3-4. A verb is missing.

P8, I5: Consider using 'events' instead of 'areas'

P8, lines 11-12, 17-20.

p10.l2: Consider deleting 'are particularly vulnerable. The sentence needs rephrasing.

p10, l29-31.

P11, l11-13.

Errors (not all errors are identified here. The article should be carefully reviewed for language):

P2. Line 18: Consider replacing 'is' with 'are'. Consider deleting 'compare' in the parentheses that refer to the Table.

P3: Line 16: 'which naturally occur exclusively in winter: the phrase is repeated after 4 sentences. Consider deleting one of these.

P.4 Line 24: ha corresponds to 'sq. m', not 'm'

Section 2.2. P.4. line 1: Consider correcting: correlation 'with' something.

p.5 line 7: consider correcting as: 'are analysed'.

p.6, line3: 'are; is missing before 'addressed'.

p.6 line 17: consider correcting ' the investigation area berlin'

---

## Referee Comment (RC2) · O. Petrucci (Referee) · 9 Feb 2018

The paper "Modelling Vulnerability to Severe Weather" exploits a dataset of fire brigade callouts in the metropolitan area of Berlin for the period of 2002-2012 to identify factors describing the local vulnerability and thus influencing the local risk for weather impacts.

The paper is quite interesting and the research was well planned. The statistical elaborations of data are carried out in a rigorous way. The language is appropriate and fluent, except for some small imperfections listed in the following. Nevertheless, there are some points that could be improved:

1) I think that the paper should have a more appropriate title. Maybe the title should focus on the statistical model that is the basis of the research, more than on a concept as wide as vulnerability.

[Figure]

2) The types of phenomena described by keywords could be more delineated by the Author. The sense of "Traffic obstruction", i.e., is not completely clear for me. In addition, a more detailed explanation of the sense of keywords could help the reader to understand the meaning of the different association of phenomena and their seasonality.

3) Personally, I think that "building density" is less significant than building coverage, and the analysis of this last one simply complicated the discussion without a clear usefulness. Nevertheless, this is a personal opinion and maybe I did not realised the actual importance of this parameter.

4) There is a point that I ask to the Author because it is not clear for me. Damage to houses/roads (as obtained by insurance data) are used together with fire brigade callouts. In the first type of data, I think that only data of damage to "goods" are included. On the contrary, I can imagine that in the fire brigade database also callouts concerning people in difficulties are included (i.e. people in cars trapped by water or by trees hurled by wind). Did you take into account this difference?

5) Because the paper seems quite long, I suggest, as general comment, to summarise some of the most important findings in a sketch or in a table. In fact, due to the multitude of variables analysed, the reader at the end of the paper is a little bit confused, even if interested.

6) I suggest to the Author to introduce, in future development of the research, data on population density. I understand that the focus is not damage to people but it must to be taken into account that population density could better approximate a sort of "total value" of the different sectors of the territory, made of people+goods that are both an object of fire brigade operations. A certain number of calls to fire brigade can be often related to dangerous situations involving people and not goods. Actually, to be strict, one should take into account the number of floor of each building and the traffic volume on the different roads, but this great amount of parameters could be very difficult to

manage. On the contrary, population density, assessed for sub sectors of the study area, is a single value, easy to asses and elaborate.

7) Legends of the figures could be simplified by putting "km per km. . ." in the caption of the figures and leaving simply the values of intervals in the figures.

8) Page 2, line 7 eliminate brackets

9) Page 4 line 10: As a first predictor, the number of houses per grid cell on a regular 1x1km grid is derived. I suggest to change houses with buildings.

10) Page 4 line 11: As discussed later, even though these quantities are highly correlated both predictors are valuable to be considered since enabling the distinction between high density city centre with very large buildings in comparison to suburban areas with high numbers of detached houses. Put comma after correlated.

11) Page 7 line 22: In general, a rather good agreement in the patterns of the number of operations per zip code area and the number of insurance claims. Please check this sentence: it seems that there is something lacking.

12) Page 8 line 3: For some events (Kyrill, Lothar07 and Aram) considerable correlation for water related operations while for the others there is no correlation at all. Please check this sentence: it seems that there is something lacking, and put a comma after events.

13) Page 8 line 6: change berlin in Berlin

14) Page 8 line 30: . . .or treefall are distributed rather different. Maybe "differently" instead that different

15) Page 8 line 35: This is not unexpected since major impacts due to treefall is not expected in wooden. Please, put a comma after unexpected.

16) Page 9 line 16: For the predictor variables listed in Table 3 the spatial. . . Please, put a comma after 3.

---

## Author Comment (AC1) · 18 Apr 2018

Thank you very much for your time reviewing our manuscript. Your comments will certainly help improving the manuscript. Please find point to point answers to your specific points ("..") below.

"The concept of the work is interesting; however there are several problems related to the methodology, the presentation of the results, the contribution of the resulting model, and consistency within the text. Furthermore, special attention to English language has to be paid, as there are several sentences that need to be corrected or rephrased."

"Major issues: 1) A major issue is the statistical methods applied and the presentation of statistical results. Nowhere in the text or Tables are any notes provided regarding the calculated correlation coefficient. What kind of correlation (pearson's r, spearman's rho...??), are data normally distributed, any tests made for that, what are the p-values

for testing significance of results. P-values are actually very important, along with the relevant significance limits (e.g. <0.05, most commonly used). The same for regression results, for which tables must provide also the standard errors, whether the coefficients are standardized, and what is the overall significance of the model (this I assume is high judging from the R2). Also, the number of observations for each correlation and regression must be provided."

Thanks for this comment. It is absolutely correct that details on the correlations as well as on the significance of results were missing in the original manuscript. Concerning the chosen method for correlation in fact we tested whether the choice (Pearson/Spearman) affected results (which it did not).We therefore used the Pearson correlation here and added reference to the methodology section. We also added significance levels (based on p-values) to results where necessary. Particularly we added significance levels to resulting correlations presented in Table 1, 2 and 3 and added the details on the procedure to the methodology section. We also modified the presentation of the regression results in Table 4 by listing regression parameters and standard deviations on the estimates and specifying the significance based on the p-values.

"2) Important methodological deficiency is related to: a) Failing to address the exact weather cause of operations and losses. The opera- tions alert keywords do not necessarily show the weather cause. This could be done only by using meteorological stations near the operations and include meteorological parameters in the analysis because these are the hazards. b) Lack of flood-specific operation or insurance data. Floods constitute the most dam- aging and expensive weather-related data. I would expect this to be clarified. Floods can cause damage to buildings, cars, trees, displacements, thus the model would have a serious value. In addition, it is not clear whether loss data have been identified as to the weather-related cause. It seems that loss data are not distributed according to the cause(s), while only windstorms/thunderstorms are addressed."

We tried to clarify on the scope of the paper, which we think was slightly misleading. We therefore added a paragraph in the introduction section which should answer part a) of this comment. The aim of this study is to identify vulnerability and exposure predictors to describe the long-term average of fire brigade operation occurrence frequencies. Therefore it is specifically not the scope to include time varying weather data (which has been done in Pardowitz and Göber (2017) and will be part of future work building more complex models combining temporally varying weather parameters and "static" vulnerability/exposure parameters. Regarding part b) of your comment: We added reference in the data section that the insurance data comprises wind-storm and hail damages to residential buildings. We agree that flooding damages are extremely valuable in addition to the analysed datasets. However data is not available.

"3) The actual contribution of the models is pending, since the predictor factors chosen ('...topographic features, land use, urban structure) are more or less constant throughout the examined period, and depict only the structure of the city. Predictors would have been the meteorological hazard parameters, which are missing from the models. I suggest the authors identify the daily weather conditions, based on meteorological data from weather stations in the city. Otherwise, the models only depict what has happened in the past based on an unchangeable 'environment'. Prediction based only on stable (endogenous) factors is not a prediction. Even the season parameter, for which a discussion was made in the methodology, is not included in the models as a dummy variable. Out of the two objectives set by the authors in the introduction (p2, lines 24-26), only the second is to my opinion reached: the identification of hotspots for weather impacts. It is not possible to 'predict the local occurrence rate of operations (p. 10, l13)' based only on these vulnerability indicators and on regressions of annual values."

This is similar to part a) of the previous comment and we think that it is due to a misunderstanding of the scope of the paper. The scope of this work is to analyze/identify predictors to describe the exposure/vulnerability to weather. This is clearly separated from the varying weather conditions on purpose! Our idea is thus based on the typical

framework used in the impact modelling community where an impact is described by an interplay of an exposure, a vulnerability and a hazard term.

In such framework we explicitly assume the exposure and vulnerability to be constant through time (even though these may also change over time which we do not consider here) while the hazard part is time varying. On the non-varying parts (exposure and vulnerability) we can base predictions of operation densities, reflecting long-term occurrence probabilities for a specific location. If I tell you -as output of the model- the mean occurrence probability for your specific neighborhood, you wouldn't call that a prediction? I guess it is a misunderstanding of what is predicted here: We aim at predicting long-term average occurrence rates for a location. And the results of our investigation in fact show that we are able to predict these long-term occurrence rates for areas in which we might not have and fire brigade operation records. Not focusing on single events this information might already be valuable information, e.g. in the planning of fire brigade capacities.

In fact these predictions have only one major shortcoming. The climate of events might be different at the location for which we make a prediction. In the current state, this is not included in the model. The model assumes a constant climate of weather events, which is certainly justified over Berlin. However it is arguable if you transfer the model to other cities or regions. In future work we plan to incorporate such effects by taking into account specific event occurrence rates (i.e. thunderstorm frequencies) varying by location.

We tried to clarify on this by adding a paragraph to the introduction on the type pf prediction we refer to. We hope this clarifies and avoids the misunderstanding we think has occurred here.

"4) Following the previous comment, there is a high correlation between the predictor factors and the outcomes. Specifically, I find it logical for the road network density to be highly correlated with traffic problems, or housing density with construction-related

impacts. This may be a methodological problem for multiple regressions. However, I can't evaluate the regressions without the required additional statistical information."

Following your first comment, we added the statistical information on the resulting parameters of the multiple regressions. We agree that it is logical that road network density correlates to traffic problems and the density of houses highly correlates with several impacts related to housing. In our study, this is exactly the information we think is valuable and we aim to harvest here. The value of knowing in how far a higher density of roads or a higher housing density affects the local volume of operations we consider a valuable information e.g. in resource planning.

One problem I believe you might be referring to is using correlated predictors in a multiple regression (multicollinearity). We added a clarifying comment on the problem of collinearity to the methods section.

"5) The authors use the expression 'potential predictors'. Then, it must be noted exactly what is the methodological contribution, because a serious problem for scholars is to find suitable and reliable predictors for relevant studies. Again, the problem is that meteorological variables or other hazard-related variables are not included or discussed in the present analysis."

As commented above, we aim at identifying predictors for vulnerability/exposure as which is why weather variables were not included here. We aim at including variable weather parameters in the future, however in our opinion this is not the scope of the paper. The word 'potential' refers to the fact that we use predictor selection to find predictors with actual (significant) contribution of skill to explain variability in the response variable (occurrence probabilities). For different operation types set of predictors with significant predictive skill are listed as an outcome of the multiple linear regression. To scholars searching for suitable predictors these predictor sets might thus be valuable starting points. We tried to modify the usage of the word potential since we think it might have been used misleadingly in the original manuscript.
"Other major issues Section 2.1. Are flood-related damages included? Please specify, since floods are the costliest weather-related hazards."

Flood-related damages are not included. We added explanations on what is included in the damage records.

"Section 2.2. Berlin-wide damages are available on daily basis for the period 1997-2011, while data on zip code level (190 within Berlin) is available for selected events only: how many events? In general, this paragraph is not clear. If only wind and lightning are included, how can the comparison with operations be made? The selection of 5 events was based on which criteria?"

The selection of events contains the 4 windstorm events with highest impacts in Berlin within the reference period 2007-2011 (Kyrill, Lothar07, Emma, Xynthia) and 2 convective events that have been selected since they were studied in Detail in Wapler et al., 2015. We added these criteria to the manuscript (specifically to the data section 2.2)

"P.4. line 1: evaluated for what exactly? Relating to the spatial distribution of operations? And why in section 4.1 you mention 6 events instead of 5?"

The events can be evaluated with respect to the spatial distribution of losses and a comparison to the occurrence of fire brigade operations. We actually analyze 6 events (4 wind storms, 2 convective events). This has been corrected in the manuscript.

"P.4. line 5: the second part of the sentence – regarding hail- is based on other references? If it is so, consider separating the sentence, because the first part refers to the analysis of losses, thus it is part of the work done by the authors."

We followed your comment and added more specific references on the winterstorm and hailfall/thunderstorm damages.

"Section 3.1. The comparison of operations and loss data is strange. As explained in section 2.2, loss data refer only to wind and thunderstorms, while operations may be related to many other causes. So, what is correlated? It is confusing."

The aim is to identify in how far specific categories of fire brigade operations (i.e. tree related operations) can be related to the wind and hail impacts as described by the insured loss data. We added clarification to section 3.1.

"Section 4.1. In general, there is an inconsistency in these correlations since flood-related losses are not included. Many assumptions have been made in the interpretation of results for this reason. E.g.: P7,l2 'coinciding with roofing damages or other wind-caused building damages' Is this an assumption? Since losses do not include flood-related damages, correlations for the water-related categories are not useful. Operations and losses are very likely to refer to different hazards."

We modified the wording in some cases to better reflect on what is an assumption/interpretation of results. The second part of your comment related on flood damages which are not included in the loss dataset. However we think it is interesting to investigate the correlation between hail damages and water related operations because for both winterstorms and thunderstorms wind, hail and precipitation may coincide. However as we added to the manuscript this also makes the interpretation of identified correlations difficulty as it is not directly clear if a correlation means that both datasets contain impacts due to the same meteorological factor (i.e. wind) or if correlations are due to the simultaneous occurrence of multiple meteorological factors. We added explanation to this in the manuscript.

"Other methodological issues: In the Abstract the building density is suggested to be the most influential predictor. However, results suggest that building coverage has the strongest effect."

We actually did not intend to suggest the building density to be the most influential predictor in the abstract. In fact both building density and building coverage are rather similar quantities and since we believed that the building density is a more commonly understood quantity we referred to this quantity in the abstract. However to be consistent we modified the abstract and refer to the building coverage.

"The expression 'water-related' operation refers to what specifically? Other than flooding, or flooding is included?"

Flooding is particularly included in this category. We added specification on the keywords and refer to an article in which details on this can be found.

"I would suggest you explain in the text what 'operations are equally distributed over winter and summer half year' mean. E.g. in a parenthesis (e.g. May to October, for summer half). Also, some local weather info would be helpful. For example, during summer half-year is there more wind and storms? And why do you make this 2-seasons choice?"

We added the information as suggested. 2 season choice is to optimally discriminate between winterstorm and thunderstorm losses. We also specified this in the manuscript.

"Why is population density not included in the predictors? What about the validation results? Why they are not shown?"

Population density is not used since (to our knowledge) there is no freely available population density dataset on a high spatial resolution (1km) as required here. Freely available datasets include the CIESIN global gridded dataset (about 5 km) or from the German Federal Statistical Office (DESTATIS) which is available on district level. Both are not sufficient for the investigation. We added reference to this in the data section.

We also added a figure exemplarily showing the validation results (see additional Figure). The Figure shows results of the iterative procedure to optimize the regression model. Increasing the penalty term for additional predictors leads to a model with smaller sets of predictor variables. For each of the resulting predictor sets and the corresponding multiple regression, the mean cross validation error (MSCVE) is calculated and plotted here. Blue circles represent validation results using a linear regression, orange circles represent results using the lognormal model and red circles represents

results using a poisson regression.

"References in the introduction regarding the use of fire service operations as impact indicator are missing. However, there are relevant papers."

We refer to a few studies in the introduction which relate emergency callout data/operation data to severe weather events. To our knowledge there are no other studies using fire service operations as weather impact indicator. We would be happy to take suggestions!

"What does a correlation between predictors and the TOTAL number of operations show, since damages can be due to different/various causes?"

This comment does not only apply to the total number of operations. Since in no case (i.e. tree-related or water-related operations) we can be certain that there is an individual meteorological cause for a certain category of impacts. By analyzing the relationships between the two impact datasets we aim at learning about the causes for various operation types (compare with our answer to your comment 2).

"Minor: Please, follow the same citation format (e.g. see differences in page 2, lines 5 and then 7."

Thanks, we made it consistent!

"Please, follow the same term for fire brigade operations. Are they callouts or operations? The authors use different terminology. However, callouts and final operations usually are 2 different things. In page 8, scetion4.2, the term used is 'operation densities'. However, it is still the number used."

We agree and avoided the term "callout" throughout the manuscript, since it might refer to something different than an actual operation. The term 'operation density' refers to the number of operations per square kilometer (which we clarified in the beginning of section 4.2)

"P.4. line 2: is it storm or thunderstorm specifically?"

Actually it is wind storm and hail damages. We clarified.

"Sections 3.2, 4.2. Consider improving the title of this section. Correlation of What (?) with potential vulnerability predictors."

We modified the title of this section to "Spatial correlation between potential vulnerability predictors and patterns of operation occurrences"

"P9, l12: 'Additional predictor variables from the CORINE land cover dataset are assessed'. Only one is shown in the discussed picture (artificial surfaces)"

We clarified that Figure 3d shows only one example for predictor variables derived from CORINE

"P10.l9: Consider removing 'considerable'. This is to be evaluated in the regression results."

We followed your comment and removed the word 'considerable'

"Sentences that should be definitely corrected/rephrased (not all cases are included. The article needs to be carefully reviewed for grammar): P2, lines 12-13. P3, lines 14-15. P4. Lines 1-2 P4. Lines 14-15 (consider changing the beginning of the sentence) P7. Lines 22-23. A verb is missing. P7, l6: 'Correlating tree-related and water-related operations'. Consider correcting as: Correlating tree-related WITH water-related operations? P8, lines 3-4. A verb is missing. P8, l5: Consider using 'events' instead of 'areas' P8, lines 11-12, 17-20. p10.l2: Consider deleting 'are particularly vulnerable. The sentence needs rephrasing. p10, l29-31. P11, l11-13.

Errors (not all errors are identified here. The article should be carefully reviewed for language): P2. Line 18: Consider replacing 'is' with 'are'. Consider deleting 'compare' in the parentheses that refer to the Table. P3: Line 16: 'which naturally occur exclusively in winter: the phrase is repeated after 4 sentences. Consider deleting one of

these. P.4 Line 24: ha corresponds to 'sq. m', not 'm' Section 2.2. P.4. line 1: Consider correcting: correlation 'with' something. p.5 line 7: consider correcting as: 'are analysed'. p.6, line3: 'are; is missing before 'addressed'. p.6 line 17: consider correcting ' the investigation area berlin'"

We thoroughly went through the sentences you mentioned and modified respectively. We also checked the whole manuscript for grammar and language.

**Fig. 1.** Additional Figure: Results of the iterative procedure to estimate the optimal predictor set (on the example of water related operations). Find additional information in the text!

---

## Author Comment (AC2) · 18 Apr 2018

Thank you very much for your time reviewing our manuscript. Your comments will certainly help improving the manuscript. Please find point to point answers to your specific points ("..") below.

"The paper "Modelling Vulnerability to Severe Weather" exploits a dataset of fire brigade callouts in the metropolitan area of Berlin for the period of 2002-2012 to identify factors describing the local vulnerability and thus influencing the local risk for weather impacts. The paper is quite interesting and the research was well planned. The statistical elab- orations of data are carried out in a rigorous way. The language is appropriate and fluent, except for some small imperfections listed in the following. Nevertheless, there are some points that could be improved: 1) I think that the paper should have a more appropriate title. Maybe the title should focus on the statistical model that is the

basis of the research, more than on a concept as wide as vulnerability."

Following your comment we propose "A Statistical Model to Estimate the Local Vulnerability to Severe Weather" as an alternative title.

"2) The types of phenomena described by keywords could be more delineated by the Author. The sense of "Traffic obstruction", i.e., is not completely clear for me. In addition, a more detailed explanation of the sense of keywords could help the reader to understand the meaning of the different association of phenomena and their seasonality."

We added details on the usage of keywords in the data section. We hope to clarify the general understanding of the keywords.

"3) Personally, I think that "building density" is less significant than building coverage, and the analysis of this last one simply complicated the discussion without a clear usefulness. Nevertheless, this is a personal opinion and maybe I did not realised the actual importance of this parameter."

It is correct that the two parameters are highly correlated. However considerable differences are present when comparing central areas of Berlin (with a dense coverage of large houses) with the outskirts of Berlin where small houses are found. While the building density (number of buildings per unit area) might be similar the coverage is much higher in the central parts of Berlin. We thus consider it worthwhile to include both as potential predictors. What we found noteworthy is the fact that in most cases building coverage is a much better predictor. However in case of tree-related operations, the correlation with the building coverage is worse, however there is a particularly high correlation with the building density. We assume that this might be due to the fact, that particularly in the outskirts of Berlin (with a high number of small buildings) the vulnerability is increased due to the presence of trees in gardens (i.e. in vicinity to buildings). We tried to clarify this in Section 4.2 without giving this discussion too much weight. "4) There is a point that I ask to the Author because it is not clear for me. Damage to houses/roads (as obtained by insurance data) are used together with fire brigade callouts. In the first type of data, I think that only data of damage to "goods" are included. On the contrary, I can imagine that in the fire brigade database also callouts concerning people in difficulties are included (i.e. people in cars trapped by water or by trees hurled by wind). Did you take into account this difference?"

We added several details to the data section (particularly 2.2. describing the building loss data) to clarify on what is contained within the insured loss data. Particularly we are aware of the fact that the different datasets contain different types of impacts. We thus do not expect to find agreement between both datasets. Instead we aim to determine differences and common features to learn about the causes for different categories of fire brigade operations. We added clarification on this to the introduction as well as Section 3.1

"5) Because the paper seems quite long, I suggest, as general comment, to summarise some of the most important findings in a sketch or in a table. In fact, due to the multitude of variables analysed, the reader at the end of the paper is a little bit confused, even if interested."

We think that Table 4 can serve as such summary since it exactly describes what you suggest. By assessing the explained variance of individual predictor variables we hope that a reader can identify the most relevant predictors. To give more weight and guide the reader we added a paragraph on this to the Conclusion chapter.

"6) I suggest to the Author to introduce, in future development of the research, data on population density. I understand that the focus is not damage to people but it must to be taken into account that population density could better approximate a sort of "total value" of the different sectors of the territory, made of people+goods that are both an object of fire brigade operations. A certain number of calls to fire brigade can be often related to dangerous situations involving people and not goods. Actually, to

СЗ

be strict, one should take into account the number of floor of each building and the traffic volume on the different roads, but this great amount of parameters could be very difficult to manage. On the contrary, population density, assessed for sub sectors of the study area, is a single value, easy to asses and elaborate."

We agree to this point. However population density is not used since (to our knowledge) there is no freely available population density dataset on a high spatial resolution (1km) as required here. Freely available datasets include the CIESIN global gridded dataset (about 5 km) or from the German Federal Statistical Office (DESTATIS) which is available on district level. Both are not sufficient for the investigation. We added reference to this in the data section. We commented on that in Section 2.3.

"7) Legends of the figures could be simplified by putting "km per km. . ." in the caption of the figures and leaving simply the values of intervals in the figures."

We will try to modify the figures to be well readable!

"8) Page 2, line 7 eliminate brackets"

Done

"9) Page 4 line 10: As a first predictor, the number of houses per grid cell on a regular 1x1km grid is derived. I suggest to change houses with buildings."

Done

"10) Page 4 line 11: As discussed later, even though these quantities are highly correlated both predictors are valuable to be considered since enabling the distinction between high density city centre with very large buildings in comparison to suburban areas with high numbers of detached houses. Put comma after correlated."

**Done**

"11) Page 7 line 22: In general, a rather good agreement in the patterns of the number of operations per zip code area and the number of insurance claims. Please check this

sentence: it seems that there is something lacking."

A verb was missing and is now added to the sentence.

"12) Page 8 line 3: For some events (Kyrill, Lothar07 and Aram) considerable correlation for water related operations while for the others there is no correlation at all. Please check this sentence: it seems that there is something lacking, and put a comma after events."

A verb was missing and is now added to the sentence. A comma was added after events.

"13) Page 8 line 6: change berlin in Berlin"

Done

"14) Page 8 line 30: ...or treefall are distributed rather different. Maybe "differently" instead that different"

Modified accordingly

"15) Page 8 line 35: This is not unexpected since major impacts due to treefall is not expected in wooden. Please, put a comma after unexpected."

Done

"16) Page 9 line 16: For the predictor variables listed in Table 3 the spatial. . . Please, put a comma after 3."

**Done**

---

## Referee Report (RR1)

The document is improved significantly and all the responses to the review were well documented. There are still points in the text where the fire brigade operations are referred as callouts (e.g. in Figure 1). The author should consider correct for consistency.

Other minor comments:

p4,l31: consider adding a comma before 'an evaluation'

p5,l25 and p9,l7 : consider adding a comma before 'which'